# Research on battery SOC estimation method by combining optimization algorithm and multi-model Kalman filtering

Zhi Ming Chen
*College of Science*
*Liaoning University of Technology*
Jinzhou, China
chenzhiminglab@163.com

Chang Qi Zhu
Navigation College
*Dalian Maritime University*
Dalian, China
zhuchangqi_work@163.com

Lei Liu
*College of Science*
*Liaoning University of Technology*
Jinzhou, China
liuleill@live.cn

*Abstract*—**With the rapid growth of electric vehicles and energy storage systems, accurate state of charge (SOC) estimation has become a critical component of battery management systems (BMS), essential for preventing overcharging and over-discharging, enhancing operational safety, and extending battery life. This paper proposes a novel SOC estimation method based on an enhanced self-correcting (ESC) model incorporating a second-order RC circuit, enabling a more accurate simulation of battery response time and dynamic behavior. To improve model reliability, a genetic algorithm-particle swarm optimization (GA-PSO) approach is employed for parameter identification. Additionally, a multi-model adaptive extended Kalman filter (AEKF) algorithm is introduced to achieve precise SOC estimation. MATLAB simulations using constant current discharge and automotive driving cycle data demonstrate that the proposed method outperforms traditional AEKF algorithms, with faster convergence and higher estimation accuracy, particularly in scenarios with varying initial estimation accuracies. The results highlight the potential of this approach to significantly enhance SOC estimation in BMS, contributing to safer operation and prolonged battery life in electric vehicles and energy storage systems.**

*Keywords—SOC estimation, Enhanced Self-Correcting model, parameter identification, GA-PSO, multi-model AEKF.*

## I. INTRODUCTION

As electric vehicles and energy storage systems continue to develop rapidly, the application of batteries as key energy storage devices has become increasingly widespread, highlighting the growing importance of battery management and control [1]. Within battery management systems (BMS), accurately estimating the state of charge (SOC) is a critical task [2]. Precise SOC estimation not only enables more reliable predictions of vehicle range but also improves battery utilization and helps prevent significant reductions in battery lifespan caused by overcharging or deep discharging [3]. However, the nonlinear characteristics, time-varying behavior, and electrochemical reactions of batteries make it impossible to measure their SOC directly with sensors [4]. Instead, SOC must be estimated using indirect measurements such as voltage, current, and temperature. Common SOC estimation methods

include approaches based on open-circuit voltage, coulomb counting, data-driven techniques, and model-based estimation methods [5]. Each of these approaches presents distinct advantages and disadvantages [6].

Among these methods, model-based estimation achieves a reasonable balance between accuracy, real-time performance, and computational cost by integrating the battery equivalent circuit model (ECM) with state estimation algorithms. The ECM is a key component in this approach. Previous studies have advanced SOC estimation using various models and algorithms. Li et al. [7] utilized a second-order RC model with a stochastic gradient algorithm for parameter identification and developed a multi-innovation extended Kalman filter, validated experimentally. Shi et al. [8] employed Bayesian belief networks and adaptive extended Kalman particle filtering, demonstrating enhanced convergence and accuracy.

However, these studies largely overlook the hysteresis effect in battery charging and discharging. Gregory L. Plett [9] addressed this by introducing an Enhanced Self-Correcting (ESC) model that incorporates hysteresis into the ECM. Sk Bittu et al. [10] simulated a first-order RC ESC model with an EKF algorithm for SOC estimation but found that the model struggles with complex polarization dynamics, and the EKF's performance deteriorates with significant measurement errors.

Accurate SOC estimation requires precise circuit modeling and effective algorithms. This study incorporates the hysteresis phenomenon using an ESC model with second-order RC characteristics. The GA-PSO algorithm is applied for precise identification of battery model parameters via an optimized fitness function. Additionally, a multi-model AEKF is developed, integrating an adaptive factor into the EKF to refine the gain matrix, thereby improving the capture of the model's dynamic properties. This multi-model approach reduces estimation errors and enhances the robustness, accuracy, and stability of SOC estimation. The main contributions of this paper are as follows:

1) Battery parameter estimation: An ESC model with second-order RC characteristics is used for accurate

characterization, with GA-PSO employed for parameter identification, validated through model testing.

2) Multi-model AEKF algorithm for SOC estimation: A multi-model AEKF algorithm is designed, combining adaptive parameters for process noise with a multi-model approach to improve SOC estimation accuracy.

3) Simulation comparative analysis: SOC estimation is analyzed using constant current discharge and automotive driving cycle scenarios, comparing the multi-model AEKF with the traditional AEKF, demonstrating enhanced convergence and accuracy..

## II. LITHIUM-ION BATTERY SOC ESTIMATION METHOD

This paper presents an ESC model based on a second-order RC equivalent circuit, incorporating the hysteresis phenomenon observed during battery charging and discharging. The model captures the battery's dynamic behavior, static characteristics, and hysteresis effects, as shown in Fig. 1.

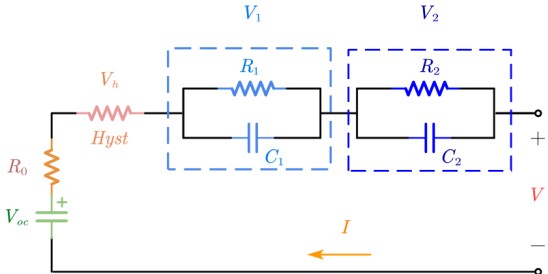

Fig. 1. ESC model of second-order RC

To identify the unknown parameters in the ESC model, the GA-PSO algorithm, an integration of Genetic Algorithm and Particle Swarm Optimization, is utilized. The process initiates with GA generating an initial population of parameter sets, which are subsequently evaluated by comparing the model's predictions with experimental battery data. GA operations, including selection, crossover, and mutation, are employed to refine these parameters, while PSO dynamically adjusts their search direction. After several iterations, the algorithm converges on the optimal parameter set, facilitating precise SOC estimation.

Building on the ESC model and parameter identification, a multi-model AEKF framework is developed to enhance SOC estimation. This framework employs multi-model fusion, integrating the estimates from several models to improve filter performance and robustness. The battery SOC is quantized into discrete sets, with n AEKF models constructed. The conditional probability of each SOC is calculated using Bayesian rules, and the SOC with the highest probability is selected for each time step. By using conditional probability as the switching rule, the multi-model AEKF adapts to varying operating conditions and improves SOC estimation accuracy and stability. The Bayesian rule used to compute these conditional probabilities is given by the following formula:

$$p(s_i|Y_k) = \frac{p(y_k|Y_{k-1},s_i)p(Y_{k-1}|s_i)p(s_i)}{\sum_{i=1}^{N} p(y_k|Y_{k-1},s_i)p(Y_{k-1}|s_i)p(s_i)} \tag{1}$$

where $p(s_i)$ denotes the prior probability, reflecting the initial estimate of the state $s_i$ in the absence of any measurement information. The entire expression delineates the posterior probability of each potential state given all previous measurements $Y_{k-1}$ and the current measurement $s_i$.

MATLAB simulations were conducted to model constant current discharge and automotive driving cycle discharge scenarios. A comparative experiment was set up between the traditional AEKF and the multi-model AEKF, focusing on evaluating their convergence performance and accuracy under conditions of unstable initial parameters and complex variations in discharge current.

## III. CONCLUSION

This paper focuses on the estimation performance of SOC in lithium-ion batteries. A second-order RC ESC model is considered, and the battery parameters are identified using the GA-PSO algorithm. Additionally, accurate estimation of battery SOC is achieved through the implementation of a multi-model Adaptive Kalman Filter. To validate the effectiveness of the proposed method, a series of simulation comparisons are conducted. The simulation results demonstrate that the proposed multi-model AEKF algorithm exhibits fast convergence and high estimation accuracy in predicting battery SOC, showcasing its superior performance in SOC estimation.

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
