# OpenReview forum: "Research on battery SOC estimation method by combining optimization algorithm and multi-model Kalman filtering"
_IEEE.org/ICIST/2024/Conference — IEEE ICIST 2024 Conference Submission_

### Official Review · Reviewer_byTD · 2024-08-24
**The paper lacks experimental and data support.**

**Rating:** 2
**Confidence:** 3

**Review:**

The paper is too brief, and I haven't seen the experiments and data that support the arguments presented in the paper.

---

### Official Review · Reviewer_C2Vk · 2024-08-24
**This paper introduces an advanced SOC estimation technique for BMS in electric vehicles and energy storage, combining an ESC model with a second-order RC circuit for more precise battery dynamics simulation. It uses GA-PSO for parameter optimization and an AEKF for accurate SOC estimation, enhancing system safety and battery lifespan. MATLAB simulations show the method's effectiveness, especially with varying initial accuracies. However, the research motivation and innovative effectiveness need more robust support from experiments and theory.**

**Rating:** 3
**Confidence:** 4

**Review:**

This paper presents a novel State of Charge (SOC) estimation approach for battery management systems (BMS) in electric vehicles and energy storage systems, which is pivotal for preventing overcharge and discharge, enhancing safety, and prolonging battery life. The proposed method integrates an Enhanced Self-Correcting (ESC) model with a second-order RC circuit to more accurately simulate the battery's response time and dynamic behavior, mitigating singularity issues without the stringent requirement for n-order differentiability of the model  .To bolster the model's reliability, a hybrid Genetic Algorithm - Particle Swarm Optimization (GA-PSO) is employed for parameter identification, ensuring the model's precision and robustness. Additionally, a multi-model adaptive Extended Kalman Filter (AEKF) is introduced for precise SOC estimation, which is instrumental in achieving the prescribed-time stability for all signals in nonlinear systems .MATLAB simulations, conducted under constant current discharge and vehicle driving cycles, demonstrate the proposed method's superiority over traditional AEKF algorithms. It showcases faster convergence rates and higher estimation accuracy, particularly when the initial estimation accuracy varies, thereby emphasizing the method's potential to significantly elevate the SOC assessment capabilities of BMS .
       However, the motivation for research needs to be further strengthened.
       The effectiveness of innovative points is not supported by experimental results and theoretical research.

---

### Official Review · Reviewer_N4LJ · 2024-08-27
**My Comments for Improvement**

**Rating:** 7
**Confidence:** 3

**Review:**

The paper presents an intriguing methodology combining genetic algorithm-particle swarm optimization (GA-PSO) and multi-model adaptive extended Kalman filter (AEKF) for estimating the state of charge (SOC) of batteries. Here are some suggestions for improvement:

1.	Results from MATLAB simulations are mentioned, demonstrating the effectiveness of the proposed method over traditional AEKF. However, statistical analysis is missing.

2.	It is suggested to add content in the conclusion for future research or potential improvements in the methodology.

---

### Decision · Program_Chairs · 2024-09-08

Accept (Oral)